# Assessment of Attractant Combinations for the Management of Red Palm Weevils (*Rhynchophorus ferrugineus*) in the United Arab Emirates

**DOI:** 10.3390/insects15040218

**Published:** 2024-03-22

**Authors:** Su-Mi Na, Gue-Il Im, Woon-Seok Lee, Dong-Gun Kim

**Affiliations:** 1Institute of Environmental Ecology, Sahmyook University, Seoul 01795, Republic of Korea; nasumi@syu.ac.kr; 2IMPROCK Co., Ltd., Bucheon 14452, Republic of Korea; miso130816@naver.com (G.-I.I.); woon@naver.com (W.-S.L.); 3Smith College of Liberal Arts, Sahmyook University, Seoul 01795, Republic of Korea

**Keywords:** attractant, ferruginol, food bait, integrated pest management, red palm weevil, pheromone

## Abstract

**Simple Summary:**

This study examined the behavior of red palm weevils in the United Arab Emirates to develop better and more environmentally friendly trapping methods. The study was conducted in three phases from June to December 2023 in Ras Al Khaimah and Abu Dhabi. Chemical attractants, such as pheromones, ethyl acetate, and food baits, like dates and coconut water, were tested. The weevils were most active between 3:00 and 6:00 a.m., with 85.72% being captured during this period. Adding pheromones to the food bait increased the capture rate by 6.95 times, and while adding food bait and ethyl acetate, and pheromones improved the capture rate by 3.14 times compared to using pheromones alone. The newly synthesized pheromone captured 2.69 times more weevils than the commercially available pheromone.

**Abstract:**

This study examined red palm weevil ecology in the United Arab Emirates to develop effective food baits, pheromone, and eco-friendly trapping methods. Three phases of investigation were conducted (from June to December 2023) on date palm farms in Ras Al Khaimah and Abu Dhabi. The first two phases, each 15 days long, were conducted in Ras Al Khaimah, whereas the third phase, 18 days long, was conducted in Abu Dhabi. Chemical attractants, such as existing pheromones and ethyl acetate, a newly synthesized ferruginol pheromone, and food baits, such as original dates, date paste, coconut water, and date palm syrup, were used to attract the weevils. Multi-funnel traps containing various attractant mixes were tested. The main activity of the red palm weevils was observed from 3:00 to 6:00 a.m., with 85.72 ± 3.39% being captured during this period, coinciding with cooler temperatures. When pheromones were added to the food bait, the capture rate increased by 6.95 ± 1.81 times. Combining food bait, ethyl acetate, and pheromones improved the capture rates by 3.14 ± 0.69 times compared to pheromones alone. The newly synthesized pheromone achieved capture rates 2.69 ± 0.07 times higher than those of the commercially available pheromone, confirming its suitability as a red palm weevil attractant.

## 1. Introduction

*Rhynchophorus* originate from Southeast Asia, India, Sri Lanka, and the Indian–Chinese Peninsula. The red palm weevil (*Rhynchophorus ferrugineus*) is the most widely distributed species of *Rhynchophorus* worldwide. In the mid-1980s, the red palm weevil was first identified as inhabiting the Middle Eastern and the Mediterranean Basins. The red palm weevil was first reported in the United Arab Emirates in Ras Al Khaimah in 1985 and it is now considered a major pest of date palms [1,2]. It later expanded to the Arabian Peninsula, Mediterranean Basin, Caribbean, and United States [3].

The red palm weevil has been reported in 50% of the countries where date palms are cultivated and 15% of countries where coconuts are cultivated [4,5]. It has been designated as a ‘Category 1’ pest of date palms in the Middle Eastern Gulf region by the Food and Agriculture Organization of the United Nations [6]. The Arabian Gulf region is responsible for approximately 30% of the global annual date palm production and suffers significant losses owing to the red palm weevil, estimated at USD 25.92 million per year [7].

The red palm weevil inhabits over 40 species of plants. However, they primarily favor 27 species, including 1 species from Agavaceae, 25 from Arecaceae, and 1 from Poaceae [8]. The red palm weevil exhibits variations in its development and reproduction depending on the host plant [9]. They are primarily attracted to wounded or dying palm trees [10] and tend to inhabit the crowns and trunks of palm trees [11]. They lay eggs primarily on damaged palm trees, but have also been observed to attack healthy palm trees [12,13,14].

Red palm weevils lay approximately 300 eggs in their lifetime [15,16]. Detection can be challenging during the early stages of infestation. However, a brownish, foul-smelling liquid oozes from the trunks of the palm trees, indicating infestation. Nonetheless, in many cases, significant damage to palm trees can occur, eventually leading to their death [11,17]. Consequently, over the past approximately 100 years, many Southeast Asian and Middle Eastern countries where date palms are cultivated have developed numerous pest control techniques. Biological methods involving the use of natural enemies have been studied to control the population of the red palm weevil [11,18,19], and studies have been conducted to utilize microorganisms to control the larvae of the red palm weevil [20,21,22]. Indeed, these methods have drawbacks such as high costs and the need for consideration of their effectiveness, especially when palm trees have already suffered damage from the red palm weevil.

To efficiently collect red palm weevils, various studies have been conducted using physical control methods such as traps, attractants, and microwaves [23,24,25,26,27,28,29]. Recently, integrated pest management (IPM) using two methods has been introduced for pest control in various countries [4,30,31,32,33]. However, several countries have refrained from using chemical control methods because of increased insecticide resistance and risks to human health [34].

The red palm weevil is distributed across approximately 52 countries [35], and thrives in various climatic zones, including tropical, Mediterranean, temperate, and desert regions [17,36]. Accordingly, manuals for the control of red palm weevils have been developed in various countries [37,38,39]. However, initial research on the red palm weevil was primarily conducted in the United States and in European countries, limiting its universal applicability due to varying climatic conditions and ecological characteristics across different countries. Furthermore, in some Southeast Asian and African regions, the larvae of the red palm weevil are raised and consumed as an alternative source of high-protein foods [40,41]. Therefore, due to the extreme climatic conditions in the Middle East, there is a pressing need for the introduction of new pest control techniques for red palm weevils.

Therefore, this study aimed to understand the ecological characteristics of red palm weevils in the United Arab Emirates, develop optimized food baits and pheromone for red palm weevils, and achieve eco-friendly and efficient pest control through trapping.

## 2. Materials and Methods

### 2.1. Study Area and Period

This study was conducted in the Ras Al Khaimah and Abu Dhabi regions of the United Arab Emirates (Figure 1). In the first and second experiments, the Ras Al Khaimah region, where the red palm weevil was first reported in the United Arab Emirates, was selected to determine food bait and identify the main activity times of the red palm weevil. The experiments were conducted for 15 days each: the first experiment was conducted from 15 June to 29 June 2023, and the second experiment was conducted from 1 July to 15 July 2023. In the Abu Dhabi region, the study was conducted for approximately 18 days, from 18 November to 5 December 2023, to verify the effectiveness of the optimal food bait and newly synthesized pheromones identified through the results of the first and second phases of the study.

### 2.2. Attractant Conditions and Monitoring

The trap used in this study was a multi-funnel trap (BioQuip, Compton, CA, USA) consisting of six stacked funnels with a total diameter of 30.6 cm and height of 71.6 cm. At the bottom, there was a cylindrical container (Figure 1c) with a diameter of 9.8 cm and height of 21 cm, designed to store food bait and capture red palm weevils. Pheromones were hung on the top of the trap, and food bait and ethyl acetate, excluding pheromones, were mixed and installed in a cylindrical container at the bottom of the trap. In the regions of Ras Al Khaimah and Abu Dhabi, three study areas were selected each, and traps were installed. The distance between the study areas was approximately 300 m; traps within the study sites were randomly placed, and the distance between the traps was 30 m. The traps were set up according to different attractant conditions. For the first and second experiments, 18 traps were installed per experiment session, with 6 per study site, and for the third experiment, a total of 15 traps were installed, with 5 traps per study site.

Fourteen combinations of attractants were used (Table 1). For the selection of the optimal food bait in the first and second experiments, the attractant compositions were based on food bait with the existing pheromone and ethyl acetate.

The food baits used original dates (Premium Dates, Natural Dates, Dubai, United Arab Emirates), date paste (Premium Dates Paste, Bayara, Dubai, United Arab Emirates), coconut water (Coconut Water Original, VITA COCO, New York, NY, USA), and date syrup (Premium Dates Syrup, Bayara, Dubai, United Arab Emirates), which is a processed product of date palm fruits. Furthermore, a small amount of mono ethylene glycol (MEG, RX Marine International, Navi Mumbai, India) was used in Type 5 to determine its impact on the attraction of red palm weevils by inhibiting moisture evaporation.

In the third experiment, to evaluate the effectiveness of the newly selected optimal food bait, two types of pheromones (existing and newly synthesized pheromones) and two types of food baits (liquid and jelly types of date paste) were utilized. The existing pheromone was Ferrolure +700 mg lure (ChemTica Internacional, Santo Domingo, Costa Rica), containing 700 mg pheromone lure (4-methyl-5-nonanol and 4-methyl-5-nonanone in a 9:1 ratio at 98% purity). The newly synthesized pheromone (IMPROCK Co., Ltd., Bucheon, Republic of Korea) was composed of a combination of the same components of the ferruginol family just like the existing pheromone, but the synthesis ratio was changed and the emission amount was set to be twice that of the existing pheromone. Gelatin powder (100% gelatin powder, HERBNARE, Seoul, Republic of Korea) was used to prepare the jelly.

The attractant composition ratio was 1:1:1 with 500 g each of water or coconut water, white sugar, and food bait, and 50 g of MEG and 300 g of gelatin powder were added. Ethyl acetate was released at 200–400 mg/day, the existing pheromone was released at 3–10 mg/day at a dose of 700 mg, and the new pheromone was released at 5–20 mg/day at a dose of 1500 mg.

For the first and second experiments, a control and five treatment groups were installed at three areas, totaling three repetitions at intervals of 5 days. To determine the main activity times of the red palm weevils, surveys were conducted at 3 h intervals. In the third experiment, a control and four treatment groups were installed in three areas. The experiment was repeated three times, with each repetition lasting 6 days. Research was conducted in 24 h intervals throughout the experiment.

### 2.3. Data Collection and Processing

Data on the number of red palm weevils were sorted according to experimental session, date, study area, attractant type, and sex. The total number of red palm weevils captured per day at each study area was calculated to be 100%, allowing the derivation of capture rates for attractants and time intervals. Temperature data were collected to determine the effect of environmental factors on the occurrence of red palm weevils. A data logger (HOBO, ONSET, 1-800-LOGGERS, Bourne, MA, USA) was installed at each study area to collect data at one-hour intervals.

### 2.4. Data Analysis and Map Production

The mean values obtained from the analysis were used as error values using the standard error. The analyses included *t*-tests, Mann–Whitney U tests, ANOVA, and Kruskal–Wallis tests. Each analysis involved performing a normality test when the sample size per group was 30 or less. If the data did not follow a normal distribution, the Mann–Whitney U and Kruskal–Wallis tests were conducted. Conversely, if the data exhibited normality, a *t*-test and ANOVA were performed. Statistical significance was determined using a significance level of α = 0.05 and a confidence level of 95%. Post hoc analysis of the ANOVA was conducted using Tukey’s test. For the Kruskal–Wallis test analysis, pairwise comparisons were performed and the Bonferroni Correction Method was used for adjustment. Statistical analyses were performed using SPSS Statistics version 25 (IBM Corp., Armonk, NY, USA) and study site maps were created using QGIS (v. 3.28.4).

## 3. Results

### 3.1. Main Catching Time

During the study period, 174 red palm weevils (67 males and 107 females) were captured. Thus, approximately 1.6 times more females were captured than males. In the first and second experiments, approximately 85.72 ± 3.39% of the total captured red palm weevils were captured between 3:00 and 6:00 a.m. Temperature analysis according to the capture times revealed that the temperature decreased after 15:00 p.m., with the lowest mean temperature being recorded at 6:00 a.m. (Figure 2). In particular, there was a noticeable pattern of a sharp temperature rise after sunrise, or after 6:00 a.m. Additionally, no red palm weevils were captured between 12:00 and 3:00 p.m., which is the period with the highest temperatures.

### 3.2. Attractant Preference

#### 3.2.1. Preference for Attractants across Experimental Sessions

In the first experiment, 35 red palm weevils were captured, comprising 17 males and 18 females. Type 2 (original dates + existing pheromone) showed the highest capture rate among the attractants, followed by Type 4 (date paste + existing pheromone) and Type 5 (date paste + existing pheromone + MEG). This indicated that attractants containing pheromones resulted in higher capture rates.

In the second experiment, 40 red palm weevils, comprising 16 males and 24 females, were captured. Among these experimental attractants, Type 7 (coconut water + existing pheromone) exhibited the highest capture rate, followed by Type 9 (date syrup + existing pheromone). Attractants containing pheromones exhibited a high overall capture rate.

In the third experiment, 99 red palm weevils were collected, comprising 34 males and 65 females. Among the attractants, Type 13 (date paste + new pheromone + liquid type) and Type 14 (date paste + new pheromone + jelly type) exhibited the highest capture rates, while Type 11 (date paste + existing pheromone + liquid type) and Type 12 (date paste + existing pheromone + jelly type) showed similar capture rates to the control group. Types 13 and 14 featured the newly synthesized pheromone, which resulted in higher capture rates than the existing pheromone (Figure 3).

#### 3.2.2. Preference for Food Baits and Pheromones

When pheromones were not included, differences in the food baits did not significantly affect the ability to attract the red palm weevils. However, when pheromones were included, the capture rate increased by a mean of 6.95 ± 1.81 times. The addition of food bait along with ethyl acetate resulted in a capture rate increase of 3.14 ± 0.69 times compared to using pheromones alone. MEG was added to prevent water loss from the attractant, but had no effect on improving the capture rates, and no statistical difference was observed between the liquid base components water and coconut water (Figure 4).

#### 3.2.3. Preference for New Pheromones

In the third experiment, the capture rates of existing and new pheromones were compared and analyzed and the capture rate of the newly synthesized attractant was high, with the capture rate increasing by 2.69 ± 0.07 times compared to the existing pheromone. Furthermore, differences in the dosage form of the attractants did not affect the attraction of red palm weevils (Figure 4).

## 4. Discussion

The ecological characteristics of the red palm weevil in the United Arab Emirates were identified and the optimal combinations of food bait and pheromones for trapping red palm weevils were determined. Weevils are diurnal insects that are primarily active during the day. However, red palm weevils from the Middle East predominantly appear during sunrise and sunset hours [29,42]. In this study, approximately 85.72 ± 3.39% of the total captured red palm weevil individuals in the United Arab Emirates region were captured between 3:00 and 6:00 a.m., indicating that the flight time of adult red palm weevils is mainly concentrated around sunrise. This trend suggests that the main activity period of the red palm weevils is in the early morning hours. This trend indicates that the high temperatures prevalent in the Middle East are limiting factors for the activity of the red palm weevil [43,44,45].

The red palm weevil pest control manual published in the Middle East recommends various pest control techniques in accordance with integrated pest management practices [46,47]. With the advancement of modern Information and Communications Technology (ICT), a technology has been developed that utilizes the sound produced when the red palm weevil larvae feed on palm trees to detect infested trees [48,49,50,51]. Additionally, the manual provides information on biological control methods, such as using microorganisms to control red palm weevil larvae [22,52].

However, all of these methods have limitations when it comes to introducing them as optimal pest control techniques after the palm trees have already been attacked, resulting in significant damage to the date palms [53,54,55,56,57]. Ethyl acetate acts as a kairomone in attracting the red palm weevil [58,59,60]. Ferruginol-based pheromones are primarily used in many countries, and the results of trapping insects using this type of pheromone have shown a higher proportion of females captured than males [57,61,62]. The new pheromone used in this study was also made from the ferruginol family. As a result of this study, a total of 174 red palm weevils were captured during the study period, of which a total of 155 individuals (60 male individuals, 95 female individuals) were captured with pheromones, with approximately 1.5 times more females being captured than males. This showed a similar trend to the results in previous research.

The capture rate using the newly synthesized pheromone was approximately 2.69 ± 0.07 times higher than that using the existing pheromone. Thus, the newly synthesized pheromone was more effective in attracting red palm weevils. In this study, the released amount of the new pheromone was 5–20 mg/day, which is twice that of the existing pheromone. According to previous studies, the capture rate of red palm weevils gradually increased when the released amount of pheromone attractant was increased to 5 mg/day, but there was no significant difference in the capture rate above 5 mg/day [59,63,64,65]. The amount of pheromone released has a direct impact on red palm weevil capture rates, suggesting that this is an essential consideration.

While the texture and type of food bait used as attractants did not affect the ability to attract red palm weevils, they play an important role as components of attractants. When components such as food bait and ethyl acetate were added to the pheromone as attractant components, the capture rate increased by 3.14 ± 0.69 times. Various studies have demonstrated the importance of fermentation products generated from bait attractants and food baits [23,31,66,67,68]. To achieve an optimal capture rate, it is important to enhance the synergy between pheromones and food baits, indicating the necessity of incorporating food baits into attractant production.

Although there was no difference in the attraction of red palm weevils based on the food bait used, when pheromones were included, the capture rate increased on mean by 6.95 ± 1.81 times. The capture rates when using original dates and coconut water were notably high. However, no statistically significant differences were observed. However, considering the economic aspects of the attractant, it is impractical to use coconut water due to its high cost. In previous studies, original dates, date stem pieces, sugarcane pieces, and the combination of original dates and ethyl acetate exhibited the best synergy [69]. Additionally, it has been reported that among food baits, original dates exhibited the highest capture rate [31]. However, this study indicated that there was no difference in the capture rates of original dates, date syrup, and date paste. According to the red palm weevil control manual, the recommended method for trapping and capturing red palm weevils involves mixing original dates in water and attaching pheromones to the top of the traps to attract the insects [37,39]. However, considering the environmental characteristics of the Middle East, where evaporation rates are high, frequent water replenishment in traps is inconvenient. In response to this, we developed jelly- and liquid-type food baits using date paste. The results showed no statistically significant differences between the two types in attracting red palm weevils. However, liquid-type bait tends to deteriorate and spoil over time. Therefore, jelly-type bait is more suitable as an attractant for the red palm weevil in the Middle East.

Utilizing the newly developed pheromone and jelly-type bait would enable more continuous and stable trapping of red palm weevils. Furthermore, the development of trapping devices incorporating ICT will enable more effective research to be conducted on the population dynamics of the red palm weevil and control measures for the red palm weevil in the United Arab Emirates.

## Figures and Tables

**Figure 1 insects-15-00218-f001:**
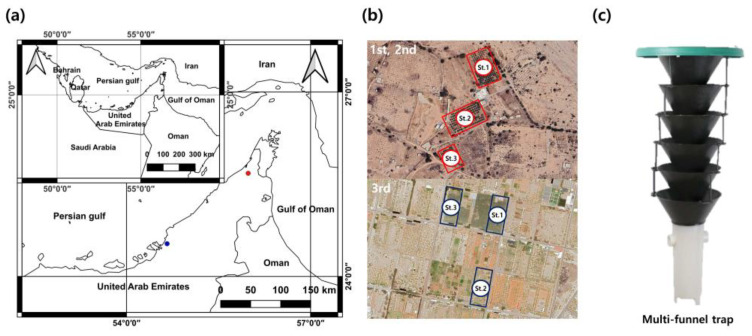
Locations of red palm weevil attractant preference experiment: (**a**,**b**) locations where the experiments were conducted and (**c**) the traps used in the experiments.

**Figure 2 insects-15-00218-f002:**
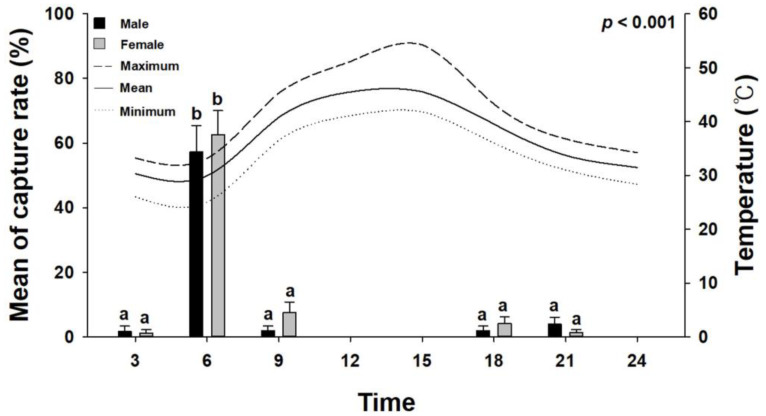
Mean capture rates of male and female red palm weevils according to collection times. These are the results of the first and second experiments conducted in Ras Al Khaimah from 15 June to 15 July 2023. The error bars represent the standard error. The letters above the standard error indicate statistical significance, indicating differences in statistical tests such as Tukey’s test, *p* < 0.001.

**Figure 3 insects-15-00218-f003:**
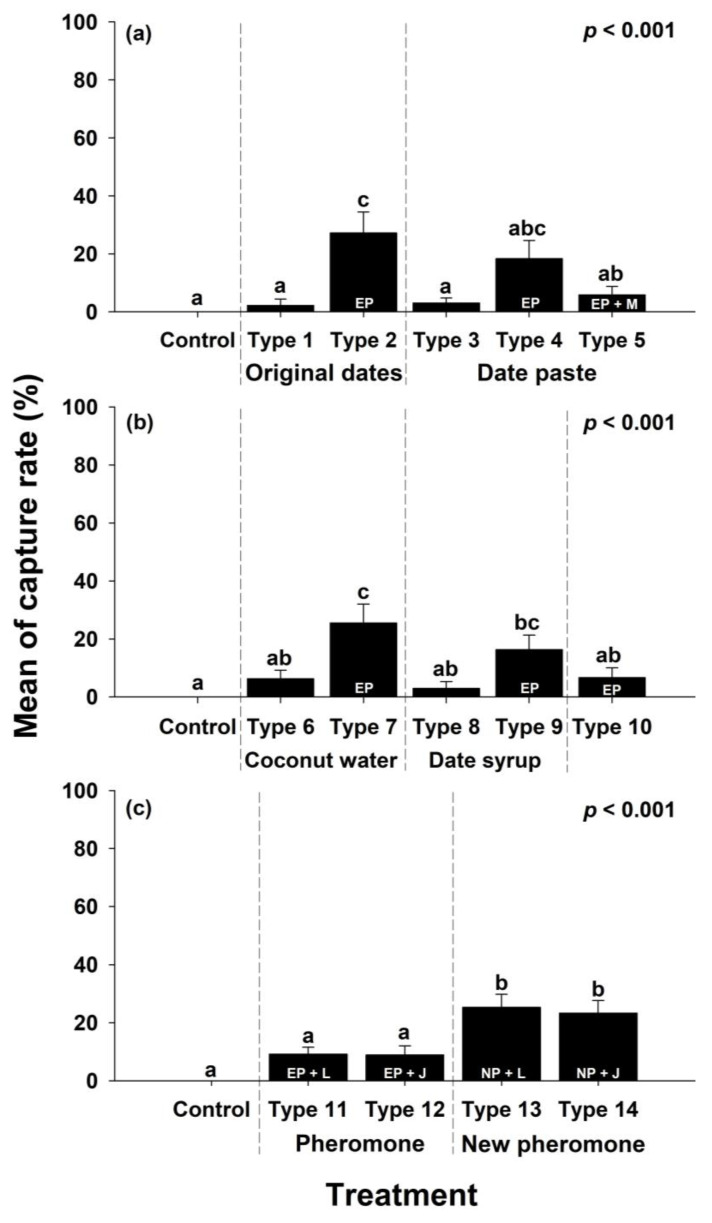
Daily mean capture rates of red palm weevils by attractant type across the experimental sessions: results of the (**a**) first, (**b**) second, and (**c**) third experiments. EP stands for existing pheromone, NP stands for new pheromone, M stands for MEG, L stands for liquid, and J stands for jelly. The error bars represent the standard error. The letters above the standard error indicate statistical significance, indicating differences in statistical tests such as Tukey’s test, *p* < 0.001.

**Figure 4 insects-15-00218-f004:**
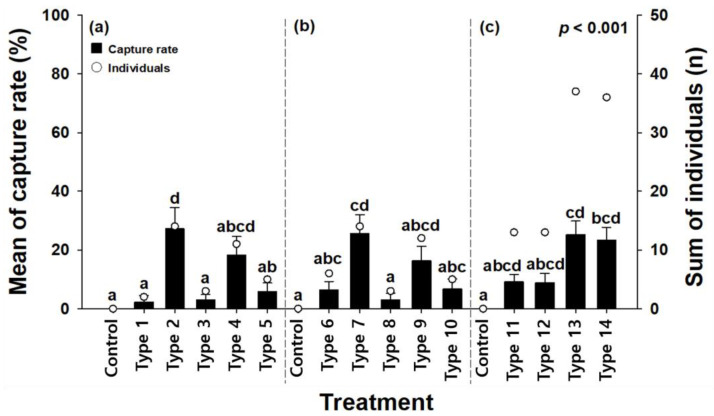
Daily mean red palm weevil capture rates by attractant type: results of the (**a**) first, (**b**) second, and (**c**) third experiments. The error bars represent the standard error. The letters above the standard error indicate statistical significance, indicating differences in statistical tests such as Tukey’s test, *p* < 0.001.

**Table 1 insects-15-00218-t001:** Attractant conditions for the red palm weevil attractant preference experiment. The first and second experiments in the United Arab Emirates’ attractant preference experiment were conducted in the Ras Al Khaimah region, the first being from 15 June to 29 June, and the second being from 1 July to 15 July; the third experiment was conducted from 18 November to 5 December in the Abu Dhabi region.

Experiment	Type	Food Baits			Pheromone	Ethyl Acetate
		Liquid Base	Additive	Dosage Form		
1st	Control	Water	-	Liquid	-	-
	Type 1	Water	White sugar + original dates	Liquid	-	○
	Type 2	Water	White sugar + original dates	Liquid	EP	○
	Type 3	Water	White sugar + date paste	Liquid	-	○
	Type 4	Water	White sugar + date paste	Liquid	EP	○
	Type 5	Water	White sugar + date paste + MEG	Liquid	EP	○
2nd	Control	Water	-	Liquid	-	-
	Type 6	Coconut water	White sugar	Liquid	-	○
	Type 7	Coconut water	White sugar	Liquid	EP	○
	Type 8	Water	White sugar + date syrup	Liquid	-	○
	Type 9	Water	White sugar + date syrup	Liquid	EP	○
	Type 10	Water	-	Liquid	EP	-
3rd	Control	Water	-	Liquid	-	-
	Type 11	Water	White sugar + date paste	Liquid	EP	○
	Type 12	Water	White sugar + date paste	Jelly	EP	○
	Type 13	Water	White sugar + date paste	Liquid	NP	○
	Type 14	Water	White sugar + date paste	Jelly	NP	○

EP: existing pheromone; NP: new pheromone; MEG: mono ethylene glycol; ○; added.

## Data Availability

The data supporting the findings of this study are available from the corresponding author upon reasonable request.

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
