# Peer review of "Assessment of Attractant Combinations for the Management of Red Palm Weevils (Rhynchophorus ferrugineus) in the United Arab Emirates"

_insects, 2024, doi:10.3390/insects15040218_

Round 1

Reviewer 1 Report

Comments and Suggestions for Authors

Assessment of Attractant Combinations for Management of 2 Red Palm Weevil (Rhynchophorus ferrugineus) in the United 3 Arab Emirates.

 Su Mi Na et al.

REVIEW

Line 21-23 – reword this. Three phases of study were done (from June to December, 2023) on date farms in etc etc….

Line 74-77 reword “Several countries have refrained from using chemical control methods because of increased insecticide resistance and riskd to human health…

FIGURE 1. all images appear to be low resolution quality on my screen and when printed off. Please check this.

Line 117 – 125 You say MEG was added to treatments to inhibit moisture evaporation, but on Table 1 it appears to be only added to type 5. MEG should probably have been added to water as an extra control measure or even by itself to clearly show it had no effect by itself. How is the “new” pheromone any different from the old pheromone? This is not explained at all apart from the statement that it was newly synthesized? Does it degrade after a certain time? What is going on there? Should all future lures be freshly made?

TABLE 1 needs to be reformatted so it is CLEAR what is in each treatment. Use clear borders for every type. I know repeating text is not great but it would be much easier to interpret this table….maybe split into three small tables.

FIGURE 2 Again I am concerned about low resolution here. Its fuzzy when printed and on screen.

Line 198 You cannot clearly say MEG had no effect by itself as your controls didn’t include it by itself. You can only say it had no effect when used synergistically with pheromone, food and water.

Line 230 where is reference citation 52?

Line 236-238 re-word this sentence.

Line 239-244 No real explanation given as to thoughts on why a newly synthesized lures works better??

Comments on the Quality of English Language

Minor editing of English language required

Author Response

Assessment of Attractant Combinations for Management of Red Palm Weevil (Rhynchophorus ferrugineus) in the United Arab Emirates.

REVIEW

  1. Line 21-23 – reword this. Three phases of study were done (from June to December, 2023) on date farms in etc etc….
  • We modified it as you suggested (Line 21-22).

  1. Line 74-77 reword “Several countries have refrained from using chemical control methods because of increased insecticide resistance and riskd to human health…
  • We modified it as you suggested (Line 75-77).

  1. FIGURE 1. all images appear to be low resolution quality on my screen and when printed off. Please check this.
  • We have uploaded the photo file as an attachment. The resolution of the attached photos is all above 600pdi.

  1. Line 117 – 125 You say MEG was added to treatments to inhibit moisture evaporation, but on Table 1 it appears to be only added to type 5. MEG should probably have been added to water as an extra control measure or even by itself to clearly show it had no effect by itself. How is the “new” pheromone any different from the old pheromone? This is not explained at all apart from the statement that it was newly synthesized? Does it degrade after a certain time? What is going on there? Should all future lures be freshly made?
  • Thank you for the kind comment. I have revised the sentence as follows (Line 127-129, 135-138, 223 – 224, 272-278).
  • MEG thought that moisture loss would affect the attraction of red palm weevil, so we created Type 5 by adding only MEG to Type 4. As a result of comparing the capture rates of Type 4 and Type 5, it was confirmed that moisture loss did not affect attraction power.
  • The new pheromone uses the same ingredients (4-methyl-5-nonanol and 4-methyl-5-nonanone) as the existing pheromone, but the synthesis ratio was different, and it was designed to emit twice the amount. Due to patent issues, we cannot disclose the exact manufacturing ratio of the new pheromone used in this study. We hope that you will understand this matter generously.
  • In the third experiment, the collection rates of existing and new pheromones were compared for 18 days, and the new pheromones showed higher capture rate and thus higher performance. The results suggest the need for changes in pheromone composition and ratio.

  1. TABLE 1 needs to be reformatted so it is CLEAR what is in each treatment. Use clear borders for every type. I know repeating text is not great but it would be much easier to interpret this table….maybe split into three small tables.
  • As you suggested, I divided with thick lines separating them according to the Experiment orders.
  • We also modified the table so that you can clearly see the ingredients.

  1. FIGURE 2 Again I am concerned about low resolution here. Its fuzzy when printed and on screen.
  • We have uploaded the photo file as an attachment. The resolution of the attached photos is all above 600pdi.

  1. Line 198 You cannot clearly say MEG had no effect by itself as your controls didn’t include it by itself. You can only say it had no effect when used synergistically with pheromone, food and water.
  • The sentence has been modified to be more specific (Line 223 – 224).
  • This sentence explains the results of type 4 and type 5. The only difference between the two attractants was whether MEG was added or not, and the rest of the ingredients were the same.
  • MEG was added because it was expected that moisture loss from the attractant would affect the capture ability of Red palm weevils.
  • However, studies have shown that water loss from the attractant did not affect the attraction of red palm weevils.
  1. Line 230 where is reference citation 52?
  • Reference citation number 52 was incorrectly listed as number 51. Due to your comment, we have rechecked the reference numbers throughout.

  1. Line 236-238 re-word this sentence.
  • The description of the newly developed attractant has been revised to be more specific (Line 272-278).

  1. Line 239-244 No real explanation given as to thoughts on why a newly synthesized lures works better??

We have added information about new synthetic pheromones in the Materials and Methods and Discussion section (Line 135-138, 272-278).

Reviewer 2 Report

Comments and Suggestions for Authors

1)     Introduction:

 physical controls using microwaves (even if still in an experimental stage) should also be included in the available means of control, and should be added in the bibliography reference;

2) Materials and methods:

A.     in the settled experimental design the number of traps used is not indicated. The number of catches of R. ferrugineus could even depend on a different number of traps per thesis;

B.     The new pheromone is not really described, therefore there is only a generic combination of pheromone family members. It is really important that in scientific experimental work, the data must be repeatable; without the knowledge of the new combination and the entirely design of the thesis, it is a non-repeatable experiment.

3)     Discussion:

 at the line 223 it’s important to highlight that the high temperatures could limiting not all the activity of the red weevil palm, but mainly the fly adults activity.

4)     Recommended bibliography:

·        Massa, R., Panariello, G., Pinchera, D., Schettino, F., Caprio, E., Griffo, R., & Migliore, M.D. - Experimental and numerical evaluations on palm microwave heating for Red Palm Weevil pest control – Nature, Scientific Reports, Published: 31 March 2017;

·        Massa, R., Schettino, F., Panariello, G., M.D. Migliore, Pinchera, D., Chirico G., D’Silva, C.J., Griffo, R., Yaseen, T. - Using microwave technology against red palm weevil: an innovative sustainable strategy to contrast a lethal pest of date palms. - ISHS Acta Horticulturae, Volume 1371, pages 71-80. 2023

Author Response

Reviewer 2

1) Introduction:

  1. physical controls using microwaves (even if still in an experimental stage) should also be included in the available means of control, and should be added in the bibliography reference;
  • As you commented, we have added content to the introduction section (Line 72-74, 375-377, 381-382).

2) Materials and methods:

  1. in the settled experimental design the number of traps used is not indicated. The number of catches of R. ferrugineus could even depend on a different number of traps per thesis;
  • We have added an explanation regarding the number of traps in the Materials and Methods section (Line 118-120).

  1. The new pheromone is not really described, therefore there is only a generic combination of pheromone family members. It is really important that in scientific experimental work, the data must be repeatable; without the knowledge of the new combination and the entirely design of the thesis, it is a non-repeatable experiment.
  • We have added information about new synthetic pheromones in the Materials and Methods section (Line 135-138).
  • The newly synthesized pheromone was composed of a combination of ingredients from the ferruginol family, just like the existing pheromones, but the synthesis ratio was different, and the emission amount was set to be twice that of the existing pheromone.

3) Discussion:

  1. at the line 223 it’s important to highlight that the high temperatures could limiting not all the activity of the red weevil palm, but mainly the fly adults activity.
  • We have modified it to reflect your comment (Line 243-246).

4) Recommended bibliography:

  • Massa, R., Panariello, G., Pinchera, D., Schettino, F., Caprio, E., Griffo, R., & Migliore, M.D. - Experimental and numerical evaluations on palm microwave heating for Red Palm Weevil pest control – Nature, Scientific Reports, Published: 31 March 2017;
  • Massa, R., Schettino, F., Panariello, G., M.D. Migliore, Pinchera, D., Chirico G., D’Silva, C.J., Griffo, R., Yaseen, T. - Using microwave technology against red palm weevil: an innovative sustainable strategy to contrast a lethal pest of date palms. - ISHS Acta Horticulturae, Volume 1371, pages 71-80. 2023
  • As you commented, we have added references to the introduction part (Line 72-74, 375-377, 381-382).

Reviewer 3 Report

Comments and Suggestions for Authors

Purpose:  This study examined the red palm weevil ecology in the United Arab Emirates, to identify the main activity times of the red palm weevil relative to air temperature and assess attractiveness of combinations of pheromones and co-attractant (ethyl acetate), food baits, and a newly synthesized pheromone.

Facts:  weevils are attracted to wounded or dying palm trees [10] and tend to inhabit the crowns and trunks of palm trees [11]. They lay eggs primarily on damaged palm trees but have been observed to attack healthy palm trees [12,13,14].

 Suggestions

Lines 24-25: reword sentence   

…such as ferruginol pheromones and co-attractant ethyl acetate, a newly synthesized ferruginol pheromone, and food baits…

Lines 61 & 62, 228:  replace infection or infected (a term noting disease) with infestation or infested.  Also, note the weevil stage (adult weevil or the immature larva) and type of feeding that is causing the oozing (larvae chew the soft tissues, gradually creating deep galleries into the tree’s structure).

Lines 72-73:  add statement and reference to known red palm weevil flight and egg laying periods in Ras Al Khaimah and Abu Dhabi.

Were weevils expected to fly and lay eggs from June 15 to July 15, 2023 and another generation fly and lay eggs from November 18 to December 5, 2023?

Lines 98-100:  add details --- how long did each experiment study period last?  and how many treatment replicates during each study period? Your text is ambiguous:

… The experiments were conducted twice, each lasting approximately one month, from June 15 to July 15, 2023, with 15 days of experimentation in each period…

Line 111: reword sentence by adding details of mg amounts in trap of each base additive

… At the bottom, there was a cylindrical container (Figure 1c) with a diameter of 9.8 cm and height of 21 cm, designed to store different treatment food baits and additives and water to retain captured red palm weevils…

Lines 111 or 121:  in either line, state where you attach the pheromone lures on either the trap top or with food bait and/or additives at bottom

Line 114:  add details of ml or mg amounts used per trap and dispenser type    ….? ml water and ?ml coconut water, with ? mg white sugar and ? ml ethyl acetate in vial (?) as additives. 

Another author (Oehlschlager 2010 in Acta horticulturae 882: 399-406) states:  …Ethyl acetate was released from plastic bottles with restrictive orifices at rate of 200-400 mg/day.

Lines 128-133:  for each of the three experimental periods, add details about trap placement above ground, trap distance to date palm plants and between traps, were traps placed in a random block design (RBD), distance between each treatment trap and replicate.

Line 134: add exact date periods per experiment and other details to Table 1 caption:  

 … attractant preference experiments in 2023 in the United Arab Emirates (1st from June 15 to July 15 and 2nd in Ras Al Khaimah region from June 15 to July 15, and 3rd in Abu Dhabi region from November 18 to December 5).

Lines 124-125: describe the composition difference between the commercial Ferrolure+® produced by ChemTica International and the newly synthesized pheromone.  State who produced the newly synthesized pheromone, is it in a liquid or jelly form, what is the lure dispensing type (sanchet, rubber septum, vial, wax plug, etc.) and note mg amount of synthesized pheromone per lure and ratio of these “same components of the ferruginol”. 

Previous authors (Abdel-Azim, Aldosari Mumtaz et al.  2017) stated more details of this weevil lure:

….The Ferrolure+[TM] sachets (Chem Tica International, Costa Rica) loaded with 700 mg of aggregation pheromone components, 4-methyl-5-nonanol and 4-methyl-5-nonanone in 9:1 ratio at 98% purity of enantiomeric mixture…

Line 158: reword sentence    … A total of 174 red palm weevils (67 males and 107 females) were captured in baited multi-funnel traps during the three experimental periods.  

Line 168: reword sentence caption noting trap site location, period(s) for recording weevil trapping data and hourly temperatures

Figure 3:   enlarge words or use enlarged abbreviations inside bars for “pheromone = P” “Liquid = L” or “Jelly = J” and describe abbreviations in Figure title

Line 221: add word   …primary activity period of the red palm weevils is in the early morning hours…

Lines 111 and 266-267:  You state:  … where evaporation rates are high, frequent water replenishment in traps is inconvenient.

Water in trap bottom to retain captured weevils may be inconvenient but state how others have retained captured insects in trap bottom capture arena without water?  Other authors (2015 J Life Science 25:12, pp.n1445-1449) use:

….Pesticide Vapor-tape II (2,2-dichlorovinyl dimethyl phosphate) (Hercon Environmental) was placed in the (funnel) trap to prevent escape of the captured beetles.

Lines 280 and 283:  Note the type of research funding.   In Line 280 says, …W.S. L. conducted the experiment and secured funding…   However, Line 283 says “….received no external funding” 

Author Response

Reviewer 3

Purpose:  This study examined the red palm weevil ecology in the United Arab Emirates, to identify the main activity times of the red palm weevil relative to air temperature and assess attractiveness of combinations of pheromones and co-attractant (ethyl acetate), food baits, and a newly synthesized pheromone.

Facts:  weevils are attracted to wounded or dying palm trees [10] and tend to inhabit the crowns and trunks of palm trees [11]. They lay eggs primarily on damaged palm trees but have been observed to attack healthy palm trees [12,13,14].

Suggestions

  1. Lines 24-25: reword sentence   

…such as ferruginol pheromones and co-attractant ethyl acetate, a newly synthesized ferruginol pheromone, and food baits…

  • We have modified the content as you suggested (Line 24-25).

  1. Lines 61 & 62, 228:  replace infection or infected (a term noting disease) with infestation or infested.  Also, note the weevil stage (adult weevil or the immature larva) and type of feeding that is causing the oozing (larvae chew the soft tissues, gradually creating deep galleries into the tree’s structure).
  • We modified the words as you suggested (Line 61, 62, 254).

  1. Lines 72-73:  add statement and reference to known red palm weevil flight and egg laying periods in Ras Al Khaimah and Abu Dhabi.

Were weevils expected to fly and lay eggs from June 15 to July 15, 2023 and another generation fly and lay eggs from November 18 to December 5, 2023?

  • Previous studies have shown that the main activity period for red palm weevil is March to April. Although the period in which this study was conducted was not the main activity period of red palm weevil, many individuals were captured using the new pheromone, demonstrating its strong attraction power.
  • The following reference is a paper that studied the main activity times of Red Farm weevil. Al-Saoud, A. H. (2013). Effect of ethyl acetate and trap colour on weevil captures in red palm weevil Rhynchophorus ferrugineus (Coleoptera: Curculionidae) pheromone traps. International Journal of Tropical Insect Science, 33, 202-206.

  1. Lines 98-100:  add details --- how long did each experiment study period last?  and how many treatment replicates during each study period? Your text is ambiguous:

… The experiments were conducted twice, each lasting approximately one month, from June 15 to July 15, 2023, with 15 days of experimentation in each period…

  • The sentence has been modified to be more specific. The first and second experiment periods were separately specified (Line 95-100).

  1. Line 111: reword sentence by adding details of mg amounts in trap of each base additive

… At the bottom, there was a cylindrical container (Figure 1c) with a diameter of 9.8 cm and height of 21 cm, designed to store different treatment food baits and additives and water to retain captured red palm weevils…

  • We have accurately documented the volume of ingredients used in the production of the attractant (Line 140-144).

  1. Lines 111 or 121:  in either line, state where you attach the pheromone lures on either the trap top or with food bait and/or additives at bottom
  • We specifically rewritten the conditions for installing pheromones and bait in the trap (Line 112-114).

  1. Line 114:  add details of ml or mg amounts used per trap and dispenser type    ….? ml water and ?ml coconut water, with ? mg white sugar and ? ml ethyl acetate in vial (?) as additives. 

Another author (Oehlschlager 2010 in Acta horticulturae 882: 399-406) states:  …Ethyl acetate was released from plastic bottles with restrictive orifices at rate of 200-400 mg/day.

  • We have rewritten the specific details regarding the volume of ingredients used in the production of bait (Line 140-144).

  1. Lines 128-133:  for each of the three experimental periods, add details about trap placement above ground, trap distance to date palm plants and between traps, were traps placed in a random block design (RBD), distance between each treatment trap and replicate.
  • We have added information about trap placement conditions (Line 114-120).

  1. Line 134: add exact date periods per experiment and other details to Table 1 caption:  

 … attractant preference experiments in 2023 in the United Arab Emirates (1st from June 15 to July 15 and 2nd in Ras Al Khaimah region from June 15 to July 15, and 3rd in Abu Dhabi region from November 18 to December 5).

  • We modified the content as you suggested (Line 151-155).

  1. Lines 124-125: describe the composition difference between the commercial Ferrolure+® produced by ChemTica International and the newly synthesized pheromone.  State who produced the newly synthesized pheromone, is it in a liquid or jelly form, what is the lure dispensing type (sanchet, rubber septum, vial, wax plug, etc.) and note mg amount of synthesized pheromone per lure and ratio of these “same components of the ferruginol”. 

Previous authors (Abdel-Azim, Aldosari Mumtaz et al.  2017) stated more details of this weevil lure:

….The Ferrolure+[TM] sachets (Chem Tica International, Costa Rica) loaded with 700 mg of aggregation pheromone components, 4-methyl-5-nonanol and 4-methyl-5-nonanone in 9:1 ratio at 98% purity of enantiomeric mixture…

  • A more detailed explanation of pheromones has been added (Line 135-138).
  • The new pheromone uses the same ingredients (4-methyl-5-nonanol, 4-methyl-5-nonanone) as the existing pheromone, but has a different synthesis ratio and is designed to be released in twice the amount. Due to patent issues, we cannot disclose the exact manufacturing ratios of the new pheromones used in this study. I hope you can kindly understand this issue.

  1. Line 158: reword sentence    … A total of 174 red palm weevils (67 males and 107 females) were captured in baited multi-funnel traps during the three experimental periods.  
  • We modified it as you suggested (Line 180).

  1. Line 168: reword sentence caption noting trap site location, period(s) for recording weevil trapping data and hourly temperatures
  • We have provided a more detailed explanation of the experiment (Line 191-192).

  1. Figure 3:   enlarge words or use enlarged abbreviations inside bars for “pheromone = P” “Liquid = L” or “Jelly = J” and describe abbreviations in Figure title
  • We modified the figure as suggested and added a description (Line 214-215).

  1. Line 221: add word   …primary activity period of the red palm weevils is in the early morning hours…
  • We have modified the sentence as you suggested (Line 247).

  1. Lines 111 and 266-267:  You state:  … where evaporation rates are high, frequent water replenishment in traps is inconvenient.

Water in trap bottom to retain captured weevils may be inconvenient but state how others have retained captured insects in trap bottom capture arena without water?  Other authors (2015 J Life Science 25:12, pp.n1445-1449) use:

….Pesticide Vapor-tape II (2,2-dichlorovinyl dimethyl phosphate) (Hercon Environmental) was placed in the (funnel) trap to prevent escape of the captured beetles.

  • The captured object was trapped in the trap's cylindrical container, and the trap was shaped like a funnel, so the captured object could not escape.

  1. Lines 280 and 283:  Note the type of research funding.   In Line 280 says, …W.S. L. conducted the experiment and secured funding…   However, Line 283 says “….received no external funding” 
  • We have corrected that sentence (Line 312).

Round 2

Reviewer 2 Report

Comments and Suggestions for Authors

the authors made the proposed changes to the text